# Learning Deep Latent-variable MRFs with Amortized Bethe Free Energy Minimization

**Sam Wiseman**
Toyota Technological Institute at Chicago
Chicago, IL 60637, USA
`swiseman@ttic.edu`

## Abstract

While much recent work has targeted learning deep discrete latent variable models with variational inference, this setting remains challenging, and it is often necessary to make use of potentially high-variance gradient estimators in optimizing the ELBO. As an alternative, we propose to optimize a non-ELBO objective derived from the Bethe free energy approximation to an MRF's partition function. This objective gives rise to a saddle-point learning problem, which we train inference networks to approximately optimize. The derived objective requires no sampling, and can be efficiently computed for many MRFs of interest. We evaluate the proposed approach in learning high-order neural HMMs on text, and find that it often outperforms other approximate inference schemes in terms of true held-out log likelihood. At the same time, we find that *all* the approximate inference-based approaches to learning high-order neural HMMs we consider underperform learning with exact inference by a significant margin.

## 1    Introduction

There has been much recent interest in learning deep generative models with discrete latent variables (Mnih & Gregor, 2014; Mnih & Rezende, 2016; Jang et al., 2017; Maddison et al., 2017; Kaiser et al., 2018; Lee et al., 2018, *inter alia*), especially in the case where these latent variables have structure – that is, where the interdependence between the discrete latents is modeled. Most recent work has focused on learning these models with variational inference (Jordan et al., 1999), and in particular with variational autoencoders (VAEs) (Kingma & Welling, 2014; Rezende et al., 2014).

Variational inference has a number of convenient properties, including that it involves the maximization of the evidence lower-bound (ELBO), a lower bound on the log marginal likelihood of the data. At the same time, when learning models with discrete latent variables variational inference may require the use of potentially high-variance gradient estimators, which are obtained during learning by sampling from the variational posterior; see Appendix A for an empirical investigation into the variance of various popular estimators when learning neural text HMMs with VAEs.

In this paper we investigate learning discrete latent variable models with an alternative objective to the ELBO. In particular, we propose to approximate the intractable log marginal likelihood with an objective deriving from the Bethe free energy (Bethe, 1935), a quantity which is intimately related to loopy belief propagation (LBP) (Pearl, 1986; Yedidia et al., 2001; 2003; Heskes, 2003), and which is the basis for "outer approximations" to the marginal polytope (Wainwright & Jordan, 2008). The Bethe free energy is attractive because if all the factors in the factor graph associated with the model have low *degree*, it can often be evaluated efficiently, without any need for approximation by sampling (see Section 2). Of course, requiring all factors in the factor graph to be of low degree severely limits the expressiveness of directed graphical

models. It does not, however, limit the expressiveness of markov random fields (MRFs) (i.e., undirected graphical models) as severely, since we can simply have an extremely loopy MRF, with arbitrary pairwise factors; see Figure 1 (c) and Section 2.2.

We accordingly propose to learn deep, undirected graphical models with latent variables, using a saddle-point objective that makes use of the Bethe free energy approximation to the model's partition functions. We further amortize inference by using "inference networks" (Srikumar et al., 2012; Kingma & Welling, 2014; Johnson et al., 2016; Tu & Gimpel, 2018) in optimizing the saddle-point objective. Unlike the ELBO, our objective will not form a lower bound on the log marginal likelihood, but an approximation to it. At the same time (and unlike other recent work on MRFs with a variational flavor (Kuleshov & Ermon, 2017; Li et al., 2019)), this objective can be optimized efficiently, without sampling, and in our experiments in learning neural HMMs on text it outperforms other approximate inference methods in terms of held out log likelihood. We emphasize, however, that despite the improvement observed when training with the proposed objective, in our experiments all approximate inference methods were found to significantly underperform learning with exact inference; see Section 4.3.

## 2 BACKGROUND AND NOTATION

### 2.1 MARKOV RANDOM FIELDS

Let $\mathcal{G} = (\mathcal{V} \cup \mathcal{F}, \mathcal{E})$ be a factor graph (Frey et al., 1997; Kschischang et al., 2001), with $\mathcal{V}$ the set of variable nodes, $\mathcal{F}$ the set of factor nodes, and $\mathcal{E}$ the set of undirected edges between elements of $\mathcal{V}$ and elements of $\mathcal{F}$; see Figure 1 (b) and (c) for two examples. We will refer collectively to variables in $\mathcal{V}$ that are always observed as $\mathbf{x}$, and to variables which are never observed as $\mathbf{z}$. In an MRF, the joint distribution then factorizes as

$$P(\mathbf{x}, \mathbf{z}; \boldsymbol{\theta}) = \frac{1}{Z(\boldsymbol{\theta})} \prod_\alpha \Psi_\alpha(\boldsymbol{x}_\alpha, \boldsymbol{z}_\alpha; \boldsymbol{\theta}),$$

where $\alpha$ indexes elements of $\mathcal{F}$, the potentials functions $\Psi_\alpha$ are assumed to always be positive and are parameterized by $\boldsymbol{\theta}$, and $Z(\boldsymbol{\theta})$ is the partition function, given by:

$$Z(\boldsymbol{\theta}) = \sum_{\boldsymbol{x}'} \sum_{\boldsymbol{z}'} \prod_\alpha \Psi_\alpha(\boldsymbol{x}'_\alpha, \boldsymbol{z}'_\alpha; \boldsymbol{\theta}).$$

We use the notation $\mathbf{x}_\alpha$ and $\mathbf{z}_\alpha$ to denote the subvectors of $\mathbf{x}$ and $\mathbf{z}$, respectively, that participate in the factor indexed by $\alpha$. (We will similarly denote subvectors of the realizations of $\mathbf{x}$ and $\mathbf{z}$ as $\boldsymbol{x}_\alpha$ and $\boldsymbol{z}_\alpha$). For example, if we had $\Psi_{22}(\boldsymbol{x}, \boldsymbol{z}) = \Psi_{22}(\boldsymbol{x}_{22}, \boldsymbol{z}_{22}) = \exp(x_1 \times x_3 + z_4)$, then $\boldsymbol{x}_{22} = [x_1; x_3]$ and $\boldsymbol{z}_{22} = z_4$.[1]

Marginalizing out the unobserved variables yields:

$$P(\mathbf{x}; \boldsymbol{\theta}) = \sum_{\boldsymbol{z}'} P(\boldsymbol{x}, \boldsymbol{z}'; \boldsymbol{\theta}) = \sum_{\boldsymbol{z}'} \frac{1}{Z(\boldsymbol{\theta})} \prod_\alpha \Psi_\alpha(\boldsymbol{x}_\alpha, \boldsymbol{z}'_\alpha; \boldsymbol{\theta}) = \frac{Z(\boldsymbol{x}, \boldsymbol{\theta})}{Z(\boldsymbol{\theta})},$$

where $Z(\boldsymbol{x}, \boldsymbol{\theta}) = \sum_{\boldsymbol{z}'} \prod_\alpha \Psi_\alpha(\boldsymbol{x}_\alpha, \boldsymbol{z}'_\alpha; \boldsymbol{\theta})$ is the partition function with $\boldsymbol{x}'$ "clamped" to $\boldsymbol{x}$ (that is, it is the partition function of $P(\boldsymbol{z} \,|\, \boldsymbol{x}; \theta)$). When learning a latent variable MRF, it is therefore natural to minimize

$$-\log P(\boldsymbol{x}; \boldsymbol{\theta}) = \log Z(\boldsymbol{\theta}) - \log Z(\boldsymbol{x}, \boldsymbol{\theta}) \tag{1}$$

with respect to $\boldsymbol{\theta}$.

---

[1] To minimize notation, we will also allow subvectors to be empty.

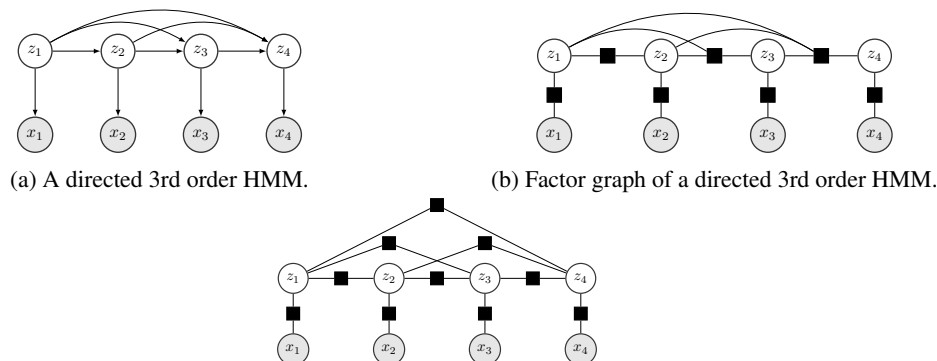

(a) A directed 3rd order HMM.

(b) Factor graph of a directed 3rd order HMM.

(c) Factor graph of an undirected HMM with only pairwise factors, but 3rd order dependencies.

Figure 1

## 2.2 BETHE FREE ENERGY

The Bethe free energy (Bethe, 1935) is a function of a parameterized factor graph and its marginals, which can be interpreted as an approximation of its clamped or unclamped partition function. To define it, first let $\tau_\alpha(\boldsymbol{x}_\alpha, \boldsymbol{z}_\alpha) \in [0, 1]$ be the marginal probability of random subvectors $\mathbf{x}_\alpha$ and $\mathbf{z}_\alpha$ taking on the values $\boldsymbol{x}_\alpha$ and $\boldsymbol{z}_\alpha$, respectively, which can be obtained by marginalizing out all variables that do not participate in the factor $\alpha$. We will refer to the vector consisting of the concatenation of the marginal probabilities for each instantiation of each factor as $\boldsymbol{\tau} \in [0, 1]^n$. As a concrete example, consider the 10 factors in Figure 1 (c): if all the $x$ variables and all the $z$ variables can take on only two possible values, then since each factor is pairwise (i.e., considers only two variables), there are $2^2$ possible settings for each factor, and thus $2^2$ corresponding marginals. In total, we then have $10 \times 4$ marginals and so $n = 40$ and $\boldsymbol{\tau} \in [0, 1]^{40}$.

Following Yedidia et al. (2003), the Bethe free energy is then defined as

$$F(\boldsymbol{\tau}) = \mathrm{KL}[q_{\boldsymbol{\tau}}(\boldsymbol{x}, \boldsymbol{z}) || P(\boldsymbol{x}, \boldsymbol{z}; \boldsymbol{\theta})] - \log Z(\boldsymbol{\theta})$$

$$= \sum_\alpha \sum_{\boldsymbol{x}'_\alpha, \boldsymbol{z}'_\alpha} \tau_\alpha(\boldsymbol{x}'_\alpha, \boldsymbol{z}'_\alpha) \log \frac{\tau_\alpha(\boldsymbol{x}'_\alpha, \boldsymbol{z}'_\alpha)}{\Psi_\alpha(\boldsymbol{x}'_\alpha, \boldsymbol{z}'_\alpha)} - \sum_{v \in \mathcal{V}} (\mathrm{ne}(x_v) - 1) \sum_{x'_v} \boldsymbol{\tau}(x'_v) \log \boldsymbol{\tau}(x'_v), \quad (2)$$

where we use the notation $q_{\boldsymbol{\tau}}(\boldsymbol{x}, \boldsymbol{z})$ to refer to a distribution with marginals $\boldsymbol{\tau}$, and where $\mathrm{ne}(x_v)$ refers to the number of factor-neighbors node $v$ has in the graphical model. The second line in equation 2 can be derived from the first by rewriting $q_{\boldsymbol{\tau}}(\boldsymbol{x}, \boldsymbol{z})$ as a product of its marginals divided by a (different) product of its marginals, which is always possible for distributions represented by tree-structured graphical models (Wainwright & Jordan, 2008), and then simplifying the resulting expressions; see Gormley & Eisner (2014) for an explicit derivation.

Crucially, since $\log Z(\boldsymbol{\theta})$ does not depend on $\boldsymbol{\tau}$, in the case of a tree-structured graphical model we have $\min_{\boldsymbol{\tau}} F(\boldsymbol{\tau}) = -\log Z(\boldsymbol{\theta})$, since the KL divergence vanishes when $\boldsymbol{\tau}$ matches the marginals of $P(\boldsymbol{x}, \boldsymbol{z}; \boldsymbol{\theta})$. In the case where the graphical model is *not* tree-structured, we may still define $F(\boldsymbol{\tau})$ as in equation 2, but in general we will only have $\min_{\boldsymbol{\tau}} F(\boldsymbol{\tau}) \approx -\log Z(\boldsymbol{\theta})$ (see Willsky et al. (2008), Weller & Jebara (2014), and Weller et al. (2014) for more precise characterizations), since the marginals $\boldsymbol{\tau}$ may not correspond to any distribution, and equation 2 no longer corresponds to a true KL divergence.

Nonetheless, the Bethe approximation may often work well in practice (Yedidia et al., 2003; Meshi et al., 2009). Indeed, Yedidia et al. (2001) show that loopy belief propagation corresponds to finding fixed points

of the constrained optimization problem $\min_{\tau \in \mathcal{C}} F(\tau)$, where $\mathcal{C}$ contains "pseudo-marginal" vectors $\tau$ that meet the following constraints: (1) each marginal distribution contained in $\tau$ consists of positive elements that sum to one, and (2) the marginals contained in $\tau$ are locally consistent in the sense that for any random variable v in the graph, and for any two factors $\alpha, \beta$ involving v, we have

$$\sum_{\boldsymbol{x}'_\alpha, \boldsymbol{z}'_\alpha : v=k} \boldsymbol{\tau}_\alpha(\boldsymbol{x}'_\alpha, \boldsymbol{z}'_\alpha) = \sum_{\boldsymbol{x}'_\beta, \boldsymbol{z}'_\beta : v=k} \boldsymbol{\tau}_\beta(\boldsymbol{x}'_\beta, \boldsymbol{z}'_\beta)$$

for all values $k$ that v might take on. In other words, the marginal probability of any random variable v taking on a particular value $k$ is consistent between all factors in which v participates. To use the variable $z_1$ in Figure 1 (c) as an example, local consistency requires that

$$\sum_{z'_2} \Psi_4(z_1 = k, z'_2) = \sum_{z'_3} \Psi_2(z_1 = k, z'_3) = \sum_{z'_4} \Psi_1(z_1 = k, z'_4) = \sum_{x'_1} \Psi_7(x'_1, z_1 = k),$$

where factors have been ordered top-down, left-to-right in the factor graph in Figure 1. Note that there are pseudo-marginals $\tau \in \mathcal{C}$ that do not correspond to the marginals of any distribution; see Wainwright & Jordan (2008) for an example.

The attractive feature of the Bethe approximation in equation 2 for our purposes is that while it is exponential in the *degree* of each factor (because it must consider every marginal), it is only linear in the number of factors. Thus, evaluating the Bethe free energy of a factor graph with a large number of small-degree (e.g., pairwise) factors remains tractable. As noted above, while this restriction severely limits the expressiveness of *directed* graphical models, MRFs are free to have arbitrary pairwise dependence. This can be seen, for example, in Figure 1, which shows two different factor graphs for a 3rd order HMM (Rabiner, 1989). While Figure 1 (b) shows the factor graph corresponding to a full directed 3rd order HMM, Figure 1 (c) shows a factor graph for a 3rd order product of experts (Hinton, 2002)-style HMM MRF, which contains only pairwise factors. Note that while Figure 1 (c) is less expressive than its directed counterpart, it still models, for instance, the dependence between $z_4$ and all the other latent variables.

## 3 AN UNSUPERVISED OBJECTIVE

We now consider learning by making use of the Bethe approximation to minimize an approximation to the log marginal likelihood of equation 1. In particular, from the previous section we have that

$$-\ln Z(\boldsymbol{\theta}) \approx \min_{\tau \in \mathcal{C}} F(\tau) \qquad \text{and} \qquad -\ln Z(\boldsymbol{x}, \boldsymbol{\theta}) \approx \min_{\tau_x \in \mathcal{C}} F_{\boldsymbol{x}}(\tau_{\boldsymbol{x}}),$$

where $F_{\boldsymbol{x}}(\tau_{\boldsymbol{x}})$ is the Bethe free energy of equation 2 with $\boldsymbol{x}$ clamped to particular values. That is, $F_{\boldsymbol{x}}$ does not consider marginals corresponding to settings of x that do not agree with $\boldsymbol{x}$, and thus $\tau_{\boldsymbol{x}}$ will in general be smaller than $\tau$. In particular, to continue the example of Figure 1 (c) from the previous section, where all variables are assumed to be binary, $\tau_{\boldsymbol{x}}$ will be in $[0, 1]^{32}$ instead of $[0, 1]^{40}$, since for each variable in $x_1, \ldots, x_4$ we ignore the two marginals corresponding to the unobserved value.

We may then define the following loss, as an approximation (but neither an upper nor lower bound) of equation 1:

$$\ell_F(\boldsymbol{\theta}) = \min_{\tau_{\boldsymbol{x}} \in \mathcal{C}} F_{\boldsymbol{x}}(\tau_{\boldsymbol{x}}) - \min_{\tau \in \mathcal{C}} F(\tau) \tag{3}$$

$$\approx -\ln Z(\boldsymbol{x}, \boldsymbol{\theta}) + \ln Z(\boldsymbol{\theta}),$$

and thus we arrive at the following saddle-point learning problem:

$$\min_{\boldsymbol{\theta}} \ell_F(\boldsymbol{\theta}) = \min_{\boldsymbol{\theta}} [\min_{\tau_{\boldsymbol{x}} \in \mathcal{C}} F_{\boldsymbol{x}}(\tau_{\boldsymbol{x}}) - \min_{\tau \in \mathcal{C}} F(\tau)]$$

$$= \min_{\boldsymbol{\theta}, \tau_{\boldsymbol{x}} \in \mathcal{C}} \max_{\tau \in \mathcal{C}} [F_{\boldsymbol{x}}(\tau_{\boldsymbol{x}}) - F(\tau)]. \tag{4}$$

## 3.1 CONSTRAINED OPTIMIZATION

While LBP can be used to find pseudo-marginals representing fixed points of $F$ and $F_x$, it is somewhat unappealing in the context of deep generative modeling, primarily because it is an iterative message-passing algorithm often requiring multiple rounds for convergence, and where the order in which messages are passed appears to significantly impact results (Yuille, 2002; Koller et al., 2009).

Instead, we propose to *predict* approximate minimizers of the Bethe free energy using trainable inference networks $f(\mathcal{G}; \boldsymbol{\phi})$ and $f_x(\mathcal{G}, \boldsymbol{x}; \boldsymbol{\phi}_x)$, which will compute approximate minimizers of $F$ and $F_x$, respectively.[2] These inference networks, which are similar to graph neural networks (Scarselli et al., 2009; Li et al., 2015; Kipf & Welling, 2016; Yoon et al., 2018) but somewhat simpler, will attempt to find minimizers living in the constraint set $\mathcal{C}$, which, as noted in Section 2.2, consists of vectors $\boldsymbol{\tau}$ containing pseudo-marginals which are positive and sum to one, and which respect local consistency. We address the parameterization of the inference networks, and how they handle these constraints on the pseudo-marginals below.

**Equality Constraints**   It is clear that the constraints that the pseudo-marginals sum to one and that they exhibit local consistency are linear constraints on $\boldsymbol{\tau}$, and so they can be expressed as $\boldsymbol{A}\boldsymbol{\tau} = \boldsymbol{b}$, where $\boldsymbol{A} \in \{0, 1\}^{m \times n}$ consists of $m$ linearly independent constraint rows and and $n$ is the length of $\boldsymbol{\tau}$. It is standard in linear-equality constrained optimization to optimize over the subspace defined by these linear constraints instead of the original optimization variable $\boldsymbol{\tau}$ (Boyd & Vandenberghe, 2004). In particular, given a feasible point $\hat{\boldsymbol{\tau}}$ and a basis $\boldsymbol{V} \in \mathbb{R}^{n \times n - m}$ for the null space of $\boldsymbol{A}$, any solution can be written as $\boldsymbol{V}\boldsymbol{u} + \hat{\boldsymbol{\tau}}$, and so we may optimize over $\boldsymbol{u}$ instead of $\boldsymbol{\tau}$.

When using inference networks to compute minimizers, however, it is more natural to have these networks output vectors of length $n$ (i.e., the size of $\boldsymbol{\tau}$, which is linear in the factors of the MRF), rather than of size $n - m$, which depends on the number of constraints. Accordingly, we may equivalently write minimizers as $\boldsymbol{V}\boldsymbol{V}^+\boldsymbol{\rho} + \hat{\boldsymbol{\tau}}$, where $\boldsymbol{\rho} \in \mathbb{R}^n$, and $\boldsymbol{V}^+$ is the Moore-Penrose pseudoinverse of $\boldsymbol{V}$. In particular, $\boldsymbol{V}\boldsymbol{V}^+$ is an orthogonal projection on the range of $\boldsymbol{V}$, a basis for the null space of $\boldsymbol{A}$. Thus, given an inference network $f$ that computes vectors in $\mathbb{R}^n$, the vector $(\boldsymbol{V}\boldsymbol{V}^+ f(\mathcal{G}; \boldsymbol{\phi}) + \hat{\boldsymbol{\tau}}) \in \mathbb{R}^n$ will satisfy the equality constraints imposed by $\mathcal{C}$, assuming $\hat{\boldsymbol{\tau}}$ does.

**Positivity Constraints**   In order to keep elements of the predicted pseudo-marginals positive, we simply impose a penalty during training on predicting a pseudo-marginal with non-positive elements. (If, during learning, a predicted pseudo-marginal is non-positive, we set it to a small positive constant). We found a linear penalty on non-positive values to work well: given a vector $\boldsymbol{\rho} \in \mathbb{R}^n$, we define the penalty function $C(\boldsymbol{\rho}) = -\frac{1}{n} \sum_{i=1}^{n} \min\{\rho_i, 0\}$. We thus arrive at the following training objective:

$$\min_{\boldsymbol{\theta}, \boldsymbol{\phi}} \max_{\boldsymbol{\phi}_x} \Big[ F_{\boldsymbol{x}}(\boldsymbol{P}_x f_x(\boldsymbol{x}, \mathcal{G}; \boldsymbol{\phi}_x) + \hat{\boldsymbol{\tau}}_x) + \lambda C(\boldsymbol{P}_x f_x(\boldsymbol{x}, \mathcal{G}; \boldsymbol{\phi}_x) + \hat{\boldsymbol{\tau}}_x) \tag{5}$$
$$- F(\boldsymbol{P} f(\mathcal{G}; \boldsymbol{\phi}) + \hat{\boldsymbol{\tau}}) - \lambda C(\boldsymbol{P} f(\mathcal{G}; \boldsymbol{\phi}) + \hat{\boldsymbol{\tau}}) \Big],$$

where $\boldsymbol{P}$ and $\boldsymbol{P}_x$ are the orthogonal projections $\boldsymbol{V}\boldsymbol{V}^+$ defined by the constraints on $\boldsymbol{\tau}$ and $\boldsymbol{\tau}_{\boldsymbol{x}}$, respectively.

## 4 EXPERIMENTS

In order to investigate how the approximate inference approach outlined above compares with other popular approaches to approximate inference, we will use approximate inference to learn in a setting in which we

---

[2]While we also experimented with obtaining approximate minimizers with gradient descent, an approach first proposed by Welling & Teh (2001) for the purpose of performing inference (but not learning), we found this approach to be both slower and less performant than using inference networks.

can tractably compute log marginal likelihoods. We consider in particular learning 2nd and 3rd order neural HMMs, as in Figure 1, on text, using various flavors of amortized variational inference as well as the Bethe-based objective $\ell_F$ introduced above. (Note that Bethe based objectives are inexact for loopy MRFs, such as the HMM-style MRFs with long-distance pairwise factors in Figure 1 (c)).

Because the Bethe objective is only interesting in the case of undirected models (e.g., Figure 1 (c)), we will strictly speaking be comparing full HMMs (learned with VAEs) with a less expressive product of expert-style variant (learned with the Bethe-based objective). However, our experiments confirm that both model classes can obtain similar held-out perplexities when learned with exact inference. We accordingly begin by briefly outlining how the neural HMMs and their associated inference networks are parameterized. To simplify the notation somewhat, in what follows we will view HMMs as parameterizing a joint distribution over a sequence of $T$ discrete observations $x_{1:T}$ and a sequence of $T$ discrete latent variables $z_{1:T}$, where each observation $z_t$ takes one of $V$ values, and each latent $z_t$ takes one of $K$ values.

## 4.1 Models

**Neural Directed HMM** We parameterize the HMM's emission distribution $P(x_t \,|\, z_t = k)$, as $\mathrm{softmax}(\boldsymbol{W} \, \mathrm{LayerNorm}(\boldsymbol{e}_k + \mathrm{MLP}(\boldsymbol{e}_k)))$, where $\boldsymbol{e}_k \in \mathbb{R}^d$ is an embedding corresponding to the $k$'th discrete value $z_t$ can take on, $\boldsymbol{W} \in \mathbb{R}^{V \times d}$ is a word embedding matrix with a row for each word in the vocabulary, and layer normalization (Ba et al., 2016) is used to stabilize training. We parameterize the transition distribution $P(z_t \,|\, z_{t-1} = k_1, \ldots, z_{t-M} = k_M)$ similarly, as $\mathrm{softmax}(\boldsymbol{U} \, \mathrm{LayerNorm}([\boldsymbol{e}_{k_1}; \ldots; \boldsymbol{e}_{k_M}] + \mathrm{MLP}([\boldsymbol{e}_{k_1}; \ldots; \boldsymbol{e}_{k_M}])))$, where $\boldsymbol{U} \in \mathbb{R}^{K \times MK}$ and the $\boldsymbol{e}_k$ are shared with the emission parameterization.

**Mean Field Style Inference Network** When performing amortized variational inference with a mean field-like posterior, we obtain approximate posteriors $q(z_t \,|\, x_{1:T})$ for each timestep $t$ as $\mathrm{softmax}(\boldsymbol{Q}\boldsymbol{h}_t)$, where $\boldsymbol{h}_t \in \mathbb{R}^{d_2}$ is the output of a bidirectional LSTM (Hochreiter & Schmidhuber, 1997; Graves et al., 2013) run over the observations $x_{1:T}$, and $\boldsymbol{Q} \in \mathbb{R}^{K \times d_2}$; note that because the bidirectional LSTM consumes all the observations this posterior is less restrictive than traditional mean field.

**Structured Inference Network** Instead of assuming the approximate posterior $q(z_{1:T} \,|\, x_{1:T})$ factorizes independently over timestep posteriors as in mean field, we can assume it is given by the posterior of a first-order (and thus more tractable) HMM. We parameterize this inference HMM identically to the neural directed HMM above, except that it conditions on the observed sequence $x_{1:T}$ by concatenating the averaged hidden states of a bidirectional LSTM run over the sequence onto the $\boldsymbol{e}_k$.

**Undirected Neural HMM** We parameterize the emission factors $\Psi_\alpha(x_t, z_t)$ as locally normalized distributions, in exactly the same way as the neural directed HMM above. In order to fairly compare with the directed HMM, the transition factors $\Psi_\alpha(z_s = k_1, z_t = k_2)$ are homogeneous (i.e., independent of the timestep), and are given by $\boldsymbol{r}_{k_2}^\mathsf{T} \mathrm{LayerNorm}([\boldsymbol{a}_{|t-s|}; \boldsymbol{e}_{k_1}] + \mathrm{MLP}([\boldsymbol{a}_{|t-s|}; \boldsymbol{e}_{k_1}]))$, where $\boldsymbol{a}_{|t-s|}$ is the embedding vector corresponding to factors relating two nodes that are $|t-s|$ steps apart, and where $\boldsymbol{e}_{k_1}$ and $\boldsymbol{r}_{k_2}$ are again discrete state embedding vectors.

**Bethe Inference Networks** In the case of sequential models like HMMs, it is fairly simple to parameterize $f(\mathcal{G}; \boldsymbol{\phi})$ and $f_x(\mathcal{G}, \boldsymbol{x}; \boldsymbol{\phi}_x)$. For $f(\mathcal{G}; \boldsymbol{\phi})$ we form an embedding for each factor $\Psi_\alpha(z_s, z_t)$ by concatenating embedding vectors corresponding to the $s$'th and $t$'th timesteps with the $K^2$ log potential values $\log \Psi_\alpha(z_s = k_1, z_t = k_2)$ (as given above), and then run a bidirectional LSTM over these factor-embeddings, ordered in ascending order of $z_s$. We then predict pseudo-marginals for each factor with a linear layer applied to the LSTM output. The parameterization of $f_x(\mathcal{G}, \boldsymbol{x}; \boldsymbol{\phi}_x)$ is similar, except we also concatenate the log observation potentials $\log \Psi_\alpha(x_s, z_s)$ and $\log \Psi_\alpha(x_t, z_t)$ onto the embedding of factor $\Psi_\alpha(z_s, z_t)$ before feeding it to the LSTM.

Table 1: Results of learning high order neural text HMMs. "Full" subtables give the performance of learning directed HMMs with a VAE objective and mean field-style ("MF") posterior approximation plus baseline ("BL"), with an IWAE objective (and $L = 5$ or $L = 10$ samples) and mean field-style posterior approximation, with a VAE objective and first order HMM ("FO") posterior approximation, and with the exact log marginal. VAE and IWAE objectives use REINFORCE-like gradient estimators. "Pairwse MRF" subtables give the performance of learning pairwise MRF HMMs with the $\ell_F$ objective but exact marginals, the $\ell_F$ objective with approximate marginals given by inference networks, and by maximizing the exact log marginal. For non-PPL objectives, we show the exponentiated, token-averaged (and negated, in the case of ELBO or IWAE) objective in order to be comparable with PPL.

| | Objective | Train PPL | Train Objective | Val. PPL | Val. Objective |
|---|---|---|---|---|---|
| | | 2nd Order HMM | | | |
| Full | MF VAE + BL | 356.278 | 7493.916 | 348.055 | 7318.822 |
| | MF IWAE, $L = 5$ | 346.241 | 7406.263 | 338.069 | 7224.112 |
| | MF IWAE, $L = 10$ | 335.622 | 7270.175 | 328.424 | 7087.252 |
| | FO HMM VAE | 279.894 | 291.144 | 286.399 | 298.078 |
| | Exact | 146.730 | N/A | 160.624 | N/A |
| Pairwise MRF | $\ell_F$ + exact marginals | 139.896 | 113.196 | 151.781 | 125.100 |
| | $\ell_F + f, f_x$ | 240.559 | 292.417 | 243.433 | 308.326 |
| | Exact | 139.491 | N/A | 149.272 | N/A |
| | | 3rd Order HMM | | | |
| Full | MF VAE + BL | 356.475 | 7512.892 | 350.983 | 7382.303 |
| | MF IWAE, $L = 5$ | 357.002 | 7545.066 | 346.573 | 7290.135 |
| | MF IWAE, $L = 10$ | 333.796 | 7251.066 | 335.241 | 7273.600 |
| | FO HMM VAE | 268.911 | 276.495 | 270.249 | 274.118 |
| | Exact | 144.889 | N/A | 159.775 | N/A |
| Pairwise MRF | $\ell_F$ + exact marginals | 159.187 | 155.038 | 170.071 | 152.881 |
| | $\ell_F + f, f_x$ | 242.063 | 293.315 | 253.345 | 254.538 |
| | Exact | 137.093 | N/A | 141.705 | N/A |

## 4.2 METHODS

We trained the HMM models described above with $K = 20$ latent states under both VAE-style and Bethe objectives on 16,737 sentences of length at most 20 from the Penn Treebank corpus (Marcus et al., 1993), and evaluated them on a held out sample of 1,585 sentences. Models and objectives were evaluated in terms of their *true* perplexity on the held out data, which can be computed reasonably efficiently with dynamic programs. We found all models and objectives to be fairly sensitive to hyperparameters and random seeds, and so we report the best results obtained (in terms of held out true perplexity) by each model and objective after a random search over hyperparameter and seed settings.

The saddle-point objective $\ell_F$ was optimized by alternating a maximization step wrt $\phi$, a minimization step wrt $\phi_x$, and a minimization step wrt $\theta$. Projection matrices $P$ for each graph structure were pre-calculated and stored, and feasible pseudo-marginals $\hat{\tau}$ can be obtained by assigning all marginals to be uniform. Code for duplicating experiments is available at https://github.com/swiseman/bethe-min, and details on hyperparameters are given in Appendix B.

### 4.3 Results and Discussion

We begin with the results obtained by maximizing the *true* log marginal likelihood of the training data under both the directed ("Full" in Table 1) and undirected models ("Pairwise MRF" in Table 1), by backpropagating gradients through the relevant dynamic programs. These results establish how well our models perform under exact inference, and are shown in the last row of each subtable in Table 1. We see that perplexities are roughly comparable between the directed and undirected models when trained with exact inference.

We now consider the remaining directed HMM results of Table 1, where the models are trained with approximate inference. In the first row of each "Full" subtable there, we show the result of maximizing the ELBO using a mean field-style posterior approximation and the REINFORCE (Williams, 1992) gradient estimator, with an input-dependent baseline to reduce variance (Mnih & Gregor, 2014). The results are quite poor, with this approximate inference scheme leading to a gain of almost 200 points in perplexity over exact inference. Using the tighter IWAE (Burda et al., 2015) objectives improves performance slightly in all cases, though the most dramatic performance improvement comes from using a first-order HMM posterior in maximizing the ELBO, which can be sampled from exactly using quantities calculated with the forward algorithm (Rabiner, 1989; Chib, 1996; Scott, 2002; Zucchini et al., 2016). While these results are encouraging, note that in general we may not have an exact dynamic program for sampling from a lower-order structured model, and that moreover we still appear to incur a perplexity penalty of more than 100 points over exact inference; see Appendix A for an empirical comparison of the variance of these estimators.

Moving to the MRF results, the second row of each "Pairwise MRF" subtable in Table 1 contains the results of optimizing $\ell_F$ as a saddle point problem. While this approach too underperforms exact inference by approximately 100 points in perplexity, somewhat remarkably it manages to consistently outperform the best approximate inference results for the directed models by a fair margin. The first row of each "Pairwise MRF" subtable in Table 1 attempts to determine whether the jump in perplexity when moving to the $\ell_F$ objective is due to the approximate inference or to the approximate objective, by minimizing the $\ell_F$ objective using the *exact* marginals, as calculated by a dynamic program. (Note that this is not equivalent to the negative log marginal likelihood, since the factor graphs are loopy). Interestingly, we see that this performs almost as well as the exact objective, suggesting that, at least for HMM models, the $\ell_F$ objective is reasonable, and approximate inference remains the problem.

Despite these encouraging results, we note that there are several drawbacks to the proposed approach. In particular, we find that in practice $\ell_F$ indeed can over- or under-estimate perplexity. Moreover, while ELBO values are not perfectly correlated with their corresponding true perplexities, values of $\ell_F$ seem even less correlated, which necessitates finding correlated proxies of perplexity that may be monitored during training. Finally, we note that explicitly calculating the projection onto the nullspace of $A$ may be prohibitive for some models (e.g., large RBMs (Smolensky, 1986)), and so other approaches to tackling the constrained optimization problem are likely necessary.

## 5 Conclusion

We have presented an objective for learning latent-variable MRFs based on the Bethe approximation to the partition function, which can often be efficiently evaluated and requires no sampling. This objective leads to slightly better held-out perplexities than other approximate inference methods when learning neural HMMs. Future work will examine scaling the proposed method to larger, non-sequential MRFs, and whether $\ell_F$-like objectives can be made to better correlate with the true perplexity.

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

## A    EMPIRICAL VARIANCE OF VARIOUS GRADIENT ESTIMATORS

Here we empirically investigate the variance of the various VAE-style gradient estimators discussed above in learning a directed 3rd-order neural HMM. In Figure 2 we plot the standard deviation of the components of the gradient with respect to the $e_k$ – the embedding vectors corresponding to each discrete latent state – averaged over all the components, as training progresses. These component-wise standard deviations are estimated from 5000 samples from the variational posterior, every 5 minibatches, for the first 350 minibatches. Figure 2 provides evidence that in practice there is indeed substantial variance associated with these estimators, and, comparing with the results in Table 1, that performance at least appears to be inversely correlated with the variance during training.

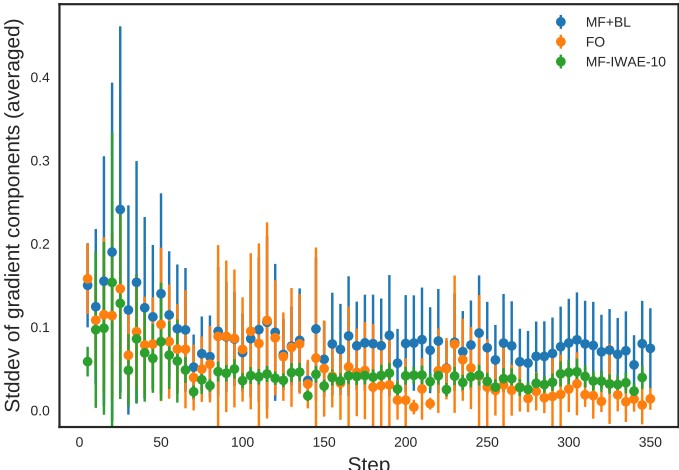

Figure 2: Average standard deviation of components of gradient with respect to $e_k$ for first 350 training steps, for the mean field-like VAE objective, the first order HMM VAE objective, and the mean field-like IWAE-10 objective; see Section 4 for details. Vertical lines are error bars.

## B    MODEL DETAILS AND HYPERPARAMETERS

Model hyperparameters are given in Table 2. All models were trained with minibatches of size 16. MRF models were trained with Adam (Kingma & Ba, 2014), while the directed models performed better with SGD.

Table 2: Best hyperparameters for various models. When hyperparameters differ between 2nd and 3rd order HMM varieties (resp.), they are separated by a comma.

| Model | Objective | $e_k$ Size | Inf. BLSTM Size | $\lambda$ |
|-------|-----------|------------|-----------------|-----------|
| Full | exact | 200, 100 | N/A | N/A |
| Full | MF VAE + BL | 100, 200 | 2x300, 2x100 | N/A |
| Full | MF IWAE-10 | 100 | 2x300 | N/A |
| Full | FO HMM VAE | 100, 300 | 2x64, 2x100 | N/A |
| MRF | exact | 200, 64 | N/A | N/A |
| MRF | $\ell_F$ | 100, 200 | 3x100 | $10^5$ |
| MRF | $\ell_F$ + exact marginals | 200 | 2x100 | N/A |

