# OpenReview forum: "Learning Deep Latent-variable MRFs with Amortized Bethe Free Energy Minimization"
_ICLR.cc/2019/Workshop/DeepGenStruct — DeepGenStruct 2019_

### Official Review · AnonReviewer1 · 2019-04-08
**Clear method for learning deep latent-variable MRFs using Bethe free energy optimization**

**Rating:** 4
**Confidence:** 2

**Review:**

This paper proposed a method for learning deep latent-variable MRF with an optimization objective that utilizes the Bethe free energy. To solve the underlying constraints of Bethe free energy optimizations, the authors proposed to represent the \tau vector using the basis of the subspace of the equality constraints and put the positivity constraints to be part of the objective. By applying these techniques, we obtain a saddle-point optimization objective with trainable inference networks and hence we can train the latent-variable MRF. The authors did some experiments on 2nd and 3rd order HMMs for empirical studies.

Pros:

1. The paper is well-written and easy to follow.

2. The original optimization for Bethe free energy is with constraints. However, the proposed objective function is without constraints, which is easier to train. The authors used the Moore-Penrose pseudoinverse of the constraint matrix V to represent the subspace of \tau, which makes the optimization process easier.

Cons and questions:

1. From the experiment results, it seems that the proposed method is not behaving well compared to the exact methods, if we do not use "exact marginals". I doubt if the performance improvements of "L_F + exact marginals" are due to the "exact marginals", not the proposed method.

2. For experiment results of the baseline methods in Table 1 and the proposed method in Table 2, the authors try to compare the PPL performances between them. Are the two experiment settings (one for directed HMMs and one for undirected HMMs) comparable? If they are, then why not putting the two tables together and compared all experiment results between them? If not, is the comparison fair?

3. It will be great if the authors can also work on some models where we can not tractably compute log marginal likelihoods, instead of only HMMs.

---

### Official Review · AnonReviewer2 · 2019-04-15
**Interesting idea**

**Rating:** 4
**Confidence:** 2

**Review:**

This paper presents an objective for learning latent variable MRFs based on Bethe free energy and amortized inference. It is different from optimizing the standard ELBO in that it does not require sampling (which has large variance) nor it is a lower/upper bound of the log-likelihood for general structured data. On some benchmark with neural HMMs, it is shown that the proposed approach achieves better held-out likelihood than other variational inference based approaches.

This paper presents an interesting idea which blends both the deep generative models research as well as the traditional Bethe free energy formulation. The prelmimarny results seems promising. I wonder how much difficult the saddle-point optimization will become on more complex models comparing with ELBO optimization.

Minor comment:

The last equation in Section 2.2: the second summation should be over z_3'.

---

### Decision · Program_Chairs · 2019-04-19
**Acceptance Decision**

Accept